:ᐧ᷀PLOS | ONE

# *In-silico* simulated prototype-patients using TPMS technology to study a potential adverse effect of sacubitril and valsartan

**Guillem Jorba**[1,2☯‡]**, Joaquim Aguirre-Plans**[2☯‡]**, Valentin Junet**[1,3]**, Cristina Segú-Vergés**[1]**, José Luis Ruiz**[1]**, Albert Pujol**[1]**, Narcís Fernández-Fuentes**[4]**, José Manuel Mas**[1]*****, Baldo Oliva**[2]*****

**1** Anaxomics Biotech SL, Barcelona, Catalonia, Spain, **2** Structural Bioinformatics Group, Research Programme on Biomedical Informatics, Department of Experimental and Health Science, Universitat Pompeu Fabra, Barcelona, Catalonia, Spain, **3** Institute of Biotechnology and Biomedicine, Universitat Autònoma de Barcelona, Cerdanyola del Vallès, Catalonia, Spain, **4** Department of Biosciences, U Science Tech, Universitat de Vic-Universitat Central de Catalunya, Vic, Catalonia, Spain

☯ These authors contributed equally to this work.
‡ These authors are joint senior authors on this work.
* baldo.oliva@upf.edu (BJ); xouse@anaxomics.com (JMM)

**Data Availability Statement:** All data and software is accessible in http://sbi.upf.edu/data/tpms

## Abstract

Unveiling the mechanism of action of a drug is key to understand the benefits and adverse reactions of a medication in an organism. However, in complex diseases such as heart diseases there is not a unique mechanism of action but a wide range of different responses depending on the patient. Exploring this collection of mechanisms is one of the clues for a future personalized medicine. The Therapeutic Performance Mapping System (TPMS) is a Systems Biology approach that generates multiple models of the mechanism of action of a drug. Each molecular mechanism generated could be associated to particular individuals, here defined as prototype-patients, hence the generation of models using TPMS technology may be used for detecting adverse effects to specific patients. TPMS operates by (1) modelling the responses in humans with an accurate description of a protein network and (2) applying a Multilayer Perceptron-like and sampling strategy to find all plausible solutions. In the present study, TPMS is applied to explore the diversity of mechanisms of action of the drug combination sacubitril/valsartan. We use TPMS to generate a wide range of models explaining the relationship between sacubitril/valsartan and heart failure (the indication), as well as evaluating their association with macular degeneration (a potential adverse effect). Among the models generated, we identify a set of mechanisms of action associated to a better response in terms of heart failure treatment, which could also be associated to macular degeneration development. Finally, a set of 30 potential biomarkers are proposed to identify mechanisms (or prototype-patients) more prone of suffering macular degeneration when presenting good heart failure response. All prototype-patients models generated are completely theoretical and therefore they do not necessarily involve clinical effects in real patients. Data and accession to software are available at http://sbi.upf.edu/data/tpms/.

**Funding:** Public funders provided support for authors salaries: JAP, NFF and BO received support from the Spanish Ministry of Economy (MINECO) [BIO2017-85329-R] [RYC-2015-17519]; "Unidad de Excelencia María de Maeztu", funded by the Spanish Ministry of Economy [ref: MDM-2014-0370]. The Research Programme on Biomedical Informatics (GRIB) is a member of the Spanish National Bioinformatics Institute (INB), PRB2-ISCIII and is supported by grant PT13/0001/0023, of the PE I+D+i 2013-2016, funded by ISCIII and FEDER. GJ has received funding from the European Union's Horizon 2020 research and innovation programme under the Marie Skłodowska-Curie grant agreement No 765912. VJ is part of a project (COSMIC; www.cosmic-h2020.eu) that has received funding from the European Union's Horizon 2020 research and innovation programme under the Marie Skłodowska-Curie grant agreement No 765158. Funding for publication is from Agència de Gestió D'ajuts Universitaris i de Recerca Generalitat de Catalunya [2017SGR00519]. The funders had no role in study design, data collection and analysis, decision to publish, or preparation of the manuscript.

**Competing interests:** I have read the journal's policy and the authors of this manuscript have the following competing interests: Baldo Oliva, currently serves on the editorial board as academic editor of PLOS ONE. The commercial affiliation (Anaxomics Biotech SL) of the authors does not alter our adherence to PLOS ONE policies on sharing data and materials

**Abbreviations:** TPMS, Therapeutic Performance Mapping System; HF, Heart Failure; MD, Macular Degeneration; MoA, Mechanism of Action; BED, Biological Effectors Database; HPN, Human Protein Network; GO, Gene Ontology.

# Introduction

Systems biology methods are an increasingly recurring strategy to understand the molecular effects of a drug in complex clinical settings [1]. Some of these methods apply computer science techniques and mathematical approaches to simulate the responses of a drug. In 2005, the Virtual Physiological Human initiative was founded with the objective of developing computational models of patients [2]. Later, they defined the concept of *In Silico* Clinical Trials as "the use of individualized computer simulation in the development or regulatory evaluation of a medicinal product, medical device, or medical intervention" [3]. Since then, *In Silico* Clinical Trials have been adopted in several occasions in preclinical and clinical trials [1].

However, current methodologies do not consider the inter-patient variability intrinsic to pharmacological treatments, missing relevant information that should be incorporated into the models. Indeed, there are many parameters influencing the Mechanisms of Action (MoA) in such therapies, including demographic data of the patient, co-treatments or clinical history. Thus, by modelling all molecular mechanisms affected by the drug, the diversity of responses observed in patients during or after the treatment could be explained.

The Therapeutic Performance Mapping System (TPMS) [4] is a method used to elucidate all the possible MoAs that could exist between an input drug and a pathology or adverse effect. It is a systems biology approach based on the simulation of patient-specific protein-protein interaction networks. TPMS incorporates data from different resources and uses the information from the drugs and diseases under study to generate multiple models of potential MoAs. In the last years, TPMS has been broadly used in different clinical areas and with different objectives [5–12], in some cases being validated in the posterior experiments [6,11,12]. Our working hypothesis is that a set of MoAs can represent the different responses to a drug in cells and that a real population of patients is the result of a myriad of cell responses. Thus, we define a prototype-patient as an abstract case with all cells responding to a single MoA.

Here, we propose the application of TPMS and protein-network approaches in the specific case study of the drug combination sacubitril/valsartan, used for the treatment of Heart Failure (HF). HF is becoming a major health problem in the western world due to its increasing hospitalization rates [13], with a prevalence being influenced by many factors like age, nutritional habits, lifestyles or genetics. This complicates the development of treatments and the identification of universal biomarkers to stratify the population. To facilitate this segmentation, it is necessary to understand the molecular details of the treatment and the pathology. Sacubitril/valsartan (marketed by Novartis as Entresto®) is a drug combination that shows better results than conventional treatments by reducing cardiovascular deaths and heart failure (HF) readmissions [14]. In pharmacological terms, it is an angiotensin receptor-neprilysin inhibitor. Consequently, it triggers the natriuretic peptide system by inhibiting neprilysin (NEP) and inhibits renin-angiotensin-aldosterone system by blocking the type-1 angiotensin II receptor (AT1R) [15]. In a previous work, TPMS was already applied to unveil the MoA of sacubitril/valsartan synergy, revealing its effect against two molecular processes [9]: the left ventricular extracellular matrix remodeling, mediated by proteins like gap junction alpha-1 protein or matrix metalloproteinase-9; and the cardiomyocyte apoptosis, through modulation of glycogen synthase kinase-3 beta. However, several publications warned about the potential long-term negative implications of using a neprilysin inhibitor like sacubitril [15–19]. Neprilysin plays a critical role at maintaining the amyloid-β homeostasis in the brain, and the alteration of amyloid-β levels has been linked to a potential long-term development of Alzheimer's disease or Macular Degeneration (MD) [15,17,19–21]. During the clinical trials PARADIGM-HF and PARAGON-HF with sacubitril/valsartan no serious effects were detected [14,22]. Still, their patient follow-up was relatively short and not specialized in finding neurodegenerative

specific symptoms. For this reason, in a forthcoming PERSPECTIVE trial (NCT02884206) a battery of cognitive tests was taken [18]. In line with this, the application of systems biology methods may shed light to the potential relationship between the treatment and the adverse effect.

In this study, we used TPMS and GUILDify v2.0 to analyze the relationship between sacubitril/valsartan, HF and MD in entirely theoretical models. Because these are theoretical models it is important to note that they are not associated with clinical effects in real patients, they only point on potential mechanisms to explain potential adverse effects. We analyzed a population of MoAs that describe the possible protein links from a sacubitril/valsartan treatment to HF and MD phenotypes. We clustered the MoAs in groups according to their response intensity and labelled them as high or low efficacy of treating HF and possibility of causing MD. We then compared these sets of MoAs and proposed a list of biomarkers to identify potential cases of MD when using sacubitril/valsartan. Simultaneously, we used GUILDify v2.0 web server [23] as an alternative approach to compare the biomarkers proposed by TPMS and reinforce the results.

## Materials and methods

### 1. Biological Effectors Database (BED) to molecularly describe specific clinical conditions

Biological Effectors Database (BED) [5,24] describes more than 300 clinical conditions as sets of genes and proteins (effectors) that can be "active", "inactive" or "neutral". For example, in a metabolic protein-like network, an enzyme will become "active" in the presence of a catalyst, or become inactivated when interacting with an inhibitor (see further details in supplementary material).

### 2. TPMS modelling

The Therapeutic Performance Mapping System (TPMS) is a tool that creates mathematical models of the protein pathways underlying a drug/pathology to explain a clinical outcome or phenotype [4–10]. These models find MoAs that explain how a *Stimulus* (i.e. proteins activated or inhibited by a drug) produces a *Response* (i.e. proteins active or inhibited in a phenotype). In the present case study, we applied TPMS to the drug-indication pair sacubitril/valsartan and HF. Regarding the drug, we retrieved the sacubitril/valsartan targets from DrugBank [25], PubChem [26], STITCH [27], SuperTarget [28] and hand curated literature revision. As for the indication, we retrieved the proteins associated with the phenotype from the BED [5,24].

**2.1. Building the Human Protein Network (HPN).** To apply the TPMS approach and create the mathematical models of MoAs, a Human Protein Network (HPN) is needed beforehand. In this study, we used a protein-protein interactions network created from the integration of public and private databases: KEGG [29], BioGRID [30], IntAct [31], REACTOME [32], TRRUST [33], and HPRD [34]. In addition, information extracted from scientific literature, which was manually curated, was also included and used for trimming the network. The resulting HPN considers interactions corresponding to different tissues to take into account the effect of the *Stimulus* in the whole body.

**2.2. Defining active/inactive nodes.** We define the state of human proteins as active or inactive for a particular phenotype, including its expression (as active) or repression (as inactive) extracted from the GSE57345 gene expression dataset [35] as in *Iborra-Egea et al* [9] (see further details in supplementary material).

**2.3. Description of the mathematical models.** The algorithm of TPMS takes as input signals the activation (+1) and inactivation (-1) of the drug target proteins, and as output the BED protein states of the pathology. It then optimizes the paths between both protein sets and computes the activation and inactivation values of all proteins in the HPN. Each node of the protein network receives as input the output of the incoming connected nodes and every link is given a weight ($\omega_l$). The sum of inputs is transformed by a hyperbolic tangent function that generates a score for every node, which becomes the "output signal" towards the outgoing connected nodes. The $\omega_l$ parameters are obtained by optimization, using a Stochastic Optimization Method based on Simulated Annealing [36]. The models are then trained by using the general restrictions (i.e. defined as edges and nodes with the property of being active or inactive) and the specific conditions set by the user. Details of the approach are shown in **Fig 1** and supplementary material.

## 3. Measures to compare sets of MoAs

To understand the relationships between all potential mechanisms we defined some measures of comparison between different sets of solutions. We expect that a drug will revert the conditions of a disease phenotype; subsequently, a drug should inactivate the active protein effectors of a pathology-phenotype and activate the inactive ones. In this section we describe the measures used in the present study to analyze and compare sets of MoAs from different views (see further details in supplementary material).

**3.1. TSignal.** To quantify the intensity of the response of a MoA, we defined TSignal as the average signal arriving at the protein effectors (equation in supplementary material).

**3.2. Distance between two sets of MoAs.** We used the modified Hausdorff distance (MHD) introduced by Dubuisson and Jain [37] as the *distance* between two or more sets of MoAs in order to determine their similarity. Details of the equations are explained in the supplementary material.

**3.3. Potential biomarkers extracted from MoAs.** In order to extract potential biomarkers when comparing sets of MoAs, we first defined the *best-classifier proteins*. These are proteins inside the HPN that allow to better classify between groups of models and are identified following a Data-Science strategy (see supplementary material). Best-classifier proteins are usually strongly related to the intensity of a response and are proteins with values differently distributed between the groups of MoAs analyzed. For this study, and for the sake of simplicity, we focused only on the 200 proteins (or pair of proteins) showing the higher classification accuracy. Assuming the hypothesis that the selected MoAs are representative of individual prototype-patients, these proteins could be used as biomarkers to classify a cohort of patients.

Then, we applied the Mann-Whitney *U* test to compare the distributions of the best-classifier proteins values between the groups and selected those proteins with significant difference (p-value< 0.01). We also restricted the list to proteins having an average value with opposite sign among groups (i.e. positive vs. negative or vice versa) and named them as *differential best-classifier proteins*. By following this strategy, we can identify two groups of differential best-classifier proteins: those active in the first group (positive output signal in average) and inactive in the other (negative output signal in average), and the opposite.

## Results and discussion

We applied TPMS to the HPN using as input signals the drug targets of sacubitril/valsartan (NEP / AT1R) and as output signals the proteins associated with HF extracted from the BED. Out of all MoAs found by TPMS, we selected the 200 satisfying the largest number of restrictions (and at least 80% of them) to perform further analysis.

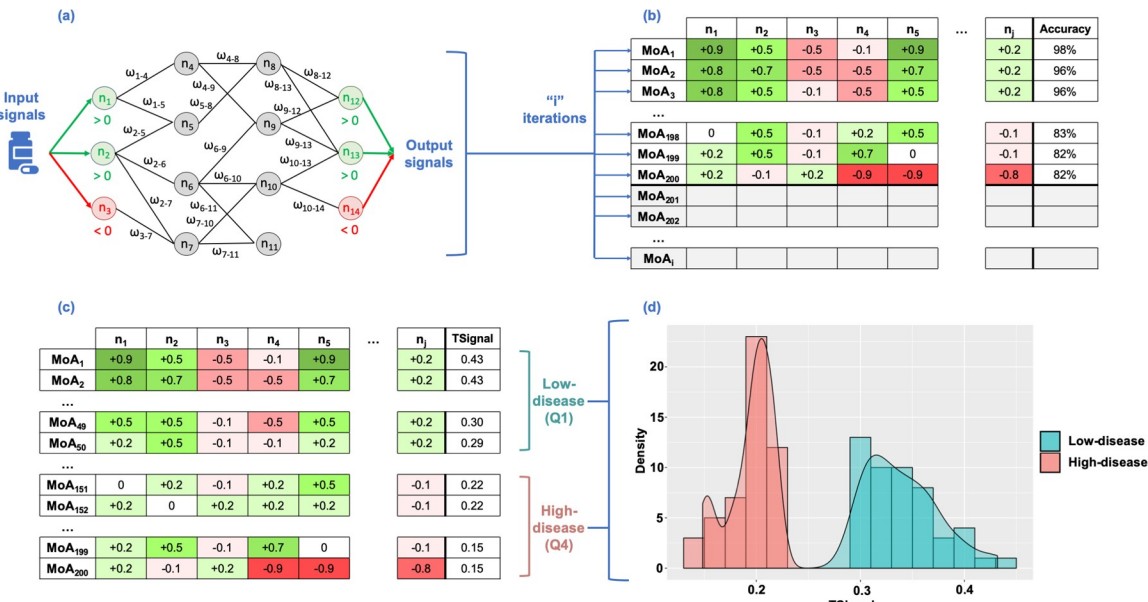

**Fig 1. Scheme of how to apply TPMS to find the Mechanisms of Action (MoA) of a drug. (a)** Scheme of the method, transmitting information over the Human Protein Network (HPN) using a Multilayer Perceptron-like and sampling. **(b)** After a given number of iterations, we obtain a collection of Mechanisms of Actions (MoA). Rows represent the MoAs and columns the output signal values of the proteins (nodes of the network). The final column shows the accuracy of the model as a percentage of the number restrictions accomplished. **(c)** 200 MoAs are selected (coloured in the slide) and sorted by TSignal. The first quartile is defined as the Low-disease group, and the fourth quartile as High-disease group. The distribution of the output signals of the two groups of MoA are shown in **(d)** (High-disease in red and Low-disease is in blue).

Note that TPMS was only executed once, optimizing the results to satisfy the restrictions on HF data. The values of MD are obtained by measuring the signal arriving at the MD effectors, which are part of the HPN and also receive signal. This procedure was chosen because we defined HF as the indication of the drug (sacubitril/valsartan), while MD is a potential adverse effect.

## 1. Stratification of MoAs

In order to compare models related to a good or bad response to the treatment, or those more prone to lead towards potential MD adverse effect, we stratified the MoAs. For HF, or treatment response, MoAs were ranked by their TSignal and then split in four quartiles. The first quartile (top 25%) contains MoAs with higher intensity of the response, which in turn corresponds to lower values of the effectors associated with HF phenotype (we named them as "Low"-disease MoAs). On the contrary, the fourth quartile (bottom 25%) collects MoAs with lower intensity of response (thus, we named as "High"-disease MoAs) (**S1 File**). On the other hand, for MD, the first quartile (top 25%) contains MoAs with higher intensity, which as an adverse event, correspond to models with high values of the effectors associated to MD (we named them as High-adverseEvent MoAs). The fourth quartile (bottom 25%) collects MoAs with lower intensity of response (thus, we named as Low- adverseEvent MoAs) (**S1 File**). Note that, in the following steps and because HF and MD groups were extracted from the same 200 set of models, common MoAs between different HF and MD-defined sets could be expected.

## 2. Comparison of MoAs with high/low TSignal associated to HF or MD

We calculated the modified Hausdorff distance between the groups of MoAs (High-MD, Low-MD, High-HF and Low-HF) to elucidate their similarity values (**S1 File**). In this sense, the

higher the distance between the groups is, the more different they are. We used these distances to calculate a dendrogram tree (see **S1 File**) showing that MoAs associated with a bad response to sacubitril/valsartan for HF (high-HF) are more similar (i.e. closer) to MoAs linked to a stronger MD adverse effect (high-MD). It is remarkable that the distances between Low- and High-HF and between Low- and High-MD are larger than the cross distances between HF and MD. However, by the definition of distance (equation 3 in supplementary material), it cannot account for the dispersion among the MoAs within and between each group. Therefore, for each set we calculated the mean Euclidean distance between all the points and its center, defined by the average of all points (see **S1 File**). As a result, all groups showed very similar dispersion values.

In order to have a global and graphical view of the distance between the individual MoAs, we generated a multidimensional scaling (MDS) plot calculated using MATLAB (see **Fig 2**). MDS plots display the pairwise distances in two dimensions while preserving the clustering characteristics (i.e. close MoAs are also close in the 2D-plot and far MoAs are also far in 2D). Focusing on the Low-HF group depicted in blue circles, we observe that there is no clear tendency to cluster with any of the MD groups. There are few cases of Low-HF MoAs coinciding in the space with Low- or High-MD MoAs. This implies that a good response to sacubitril/valsartan of HF patients would not be usually linked to the development of MD. Moreover, no clear distinction is found when plotting only the MD MoAs within the Low-HF group (see **S1 File**). However, regarding the set of High-HF MoAs, we can differentiate two clusters of MoAs: one related to the High-MD group (green crosses); and the other close to MoAs of the Low-MD group (black crosses) (see **S1 File**).

Assuming the hypothesis that different MoAs correspond to distinct prototype-patients, we conclude that for the specific set of patients for which sacubitril/valsartan works best reducing HF, it would be more difficult to differentiate between those presenting MD and those who do not. Instead, for the High-HF group, patients having MD could indeed be easily distinguished from those not presenting MD as side effect. However, because Low-HF group has more relevance to the clinics, specific functional analyses were performed in this specific group, as seen in following sections. Finally, we highlight that, as these distinct groups of prototype-patients are theoretical simulations, they don't reflect the clinical effects of real patients.

## 3. Identification and functional analysis of potential biomarkers

For this section, we identified the nodes (i.e. proteins) significantly differentiating two groups of models (using a Mann-Whitney *U* test) for which the average of output signals have opposite signs (see methods in 3.3). After that, the function of the identified proteins was extracted from Gene Ontology (GO).

**3.1. Identification of best-classifier proteins differentiating HF responses.** After comparing High- vs Low- HF groups, we found a total of 45 differential best-classifier proteins associated with the treatment response (6 Low-HF-active/High-HF-inactive and 39 Low-HF-inactive/High-HF-active) (see **Fig 3A** and **S1 File**). To pinpoint the biological role of these proteins, we first identified the GO enriched functions (see **S1 File**) and then searched in the literature for evidences linking them with HF. As a result, we found that the differential best-classifier proteins Low-HF-active/High-HF-inactive point towards an important role for actin nucleation and polymerization mechanisms in drug response (reflected by the functions *regulation of actin nucleation*, *regulation of Arp2/3 complex-mediated actin nucleation*, *SCAR complex*, *filopodium tip*, or *dendrite extension*). In fact, the alteration of actin nucleation and polymerization mechanisms has been reported in heart failure [38–40]. Interestingly, a role for the activation of another differential best-classifier candidate, ATGR2, has been proposed to

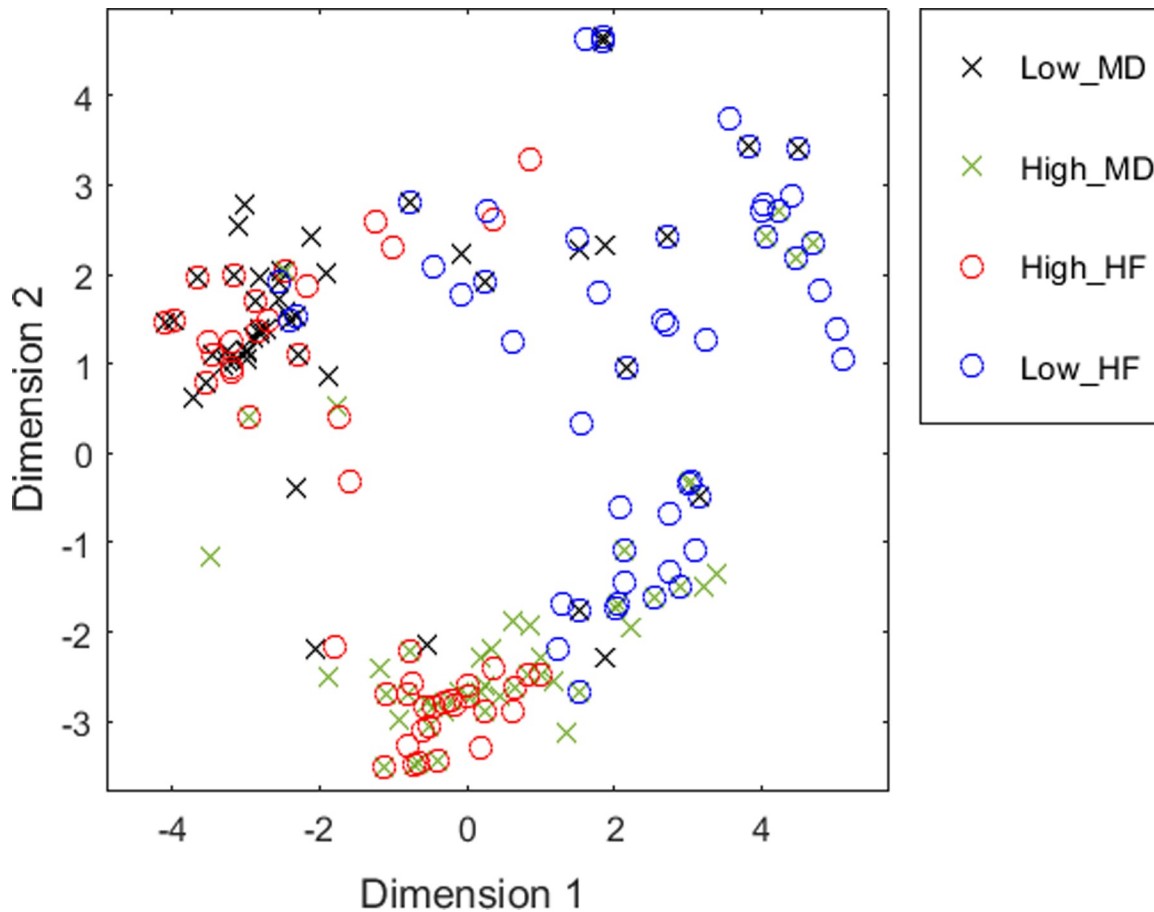

**Fig 2. Multidimensional scaling plot of the distances between the Mechanisms of Action (MoA) of the four groups defined.** Each point represents a MoA. Axes are defined by the most representative dimensions.

mediate some of the beneficial effects of angiotensin II receptor type 1 antagonists, such as valsartan [41,42]. On the other hand, the results of the differential best-classifier proteins Low-HF-inactive/High-HF-active are linked to phosphatidylinositol kinase mediated pathways (*phosphatidylinositol-3,4-bisphosphate 5-kinase activity*) and MAP kinase mediated pathways (*MAP kinase kinase activity*, best classifier proteins MAPK1, MAPK3, MAPK11, MAPK12 or MAPK13). In this case, both signaling pathways have been associated to cardiac hypertrophy and subsequent heart failure [43,44]. These outcomes clearly lead towards the idea that High-HF models are a representation of prototype-patients with a worst response to the treatment, while Low-HF models are related to more beneficial response to the medication. A more detailed explanation can be found in the supplementary material.

**3.2. Identification of best-classifier proteins differentiating MD responses.** We identified 57 differential best-classifier proteins of MD (28 Low-MD-active/High-MD-inactive and 29 Low-MD-inactive/High-MD-active) (see **Fig 3B** and **S1 File**). Again, we searched for relationships between these proteins and MD by identifying the GO enriched functions (see **S1 File**) and searching for links in the literature. Some of the proteins and functions highlighted in the current analysis had been related to MD in previous works. The presence of dendritic spine development and dorsal/ventral axon guidance related proteins emphasizes the role of sacubitril/valsartan in dendritic and synaptic plasticity mechanisms, which had been previously linked to MD [45]. Furthermore, valsartan treatment has been reported to promote

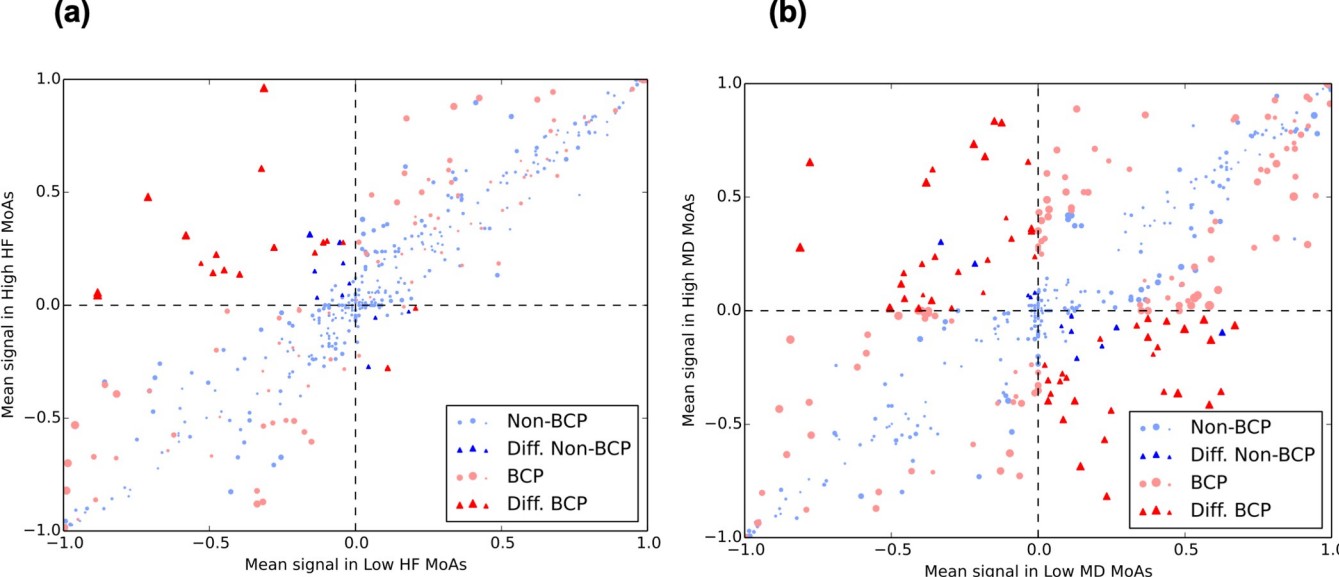

**(a)** **(b)**

**Fig 3. Scatter plot of the mean signal values of Low and High-"disease" Mechanisms of Action (MoA).** Scatter plot of the mean signal values of Low-"disease" and High-"disease" MoAs for each protein using as disease Heart Failure (HF) in **(a)** and Macular Degeneration (MD) in **(b)**. The average of the output signal of each protein in High-group is presented versus its value in Low-group. Differential signals (Diff., shown as triangles) are defined as those with opposite sign when comparing High versus Low average, and a p-value < 0.01 when calculating the Mann-Whitney *U* test between the two distributions of signals. Best-classifier proteins (BCP) are colored in red, otherwise they are blue. Sizes of markers are proportional to p-values of the Mann-Whitney *U* test.

dendritic spine development in other related neurodegenerative diseases, such as Alzheimer's disease [46]. Other enriched functions are implicated in growth factor related pathways, which are known to be involved in wet MD pathogenesis [47]. Moreover, neovascularization in the wet variant of MD has been linked to the signaling of some of the growth factors detected as sacubitril/valsartan-associated MD classifiers in this study, including FGF1 [47] and PDGF [48,49]. A more detailed explanation can be found in the supplementary material.

**3.3. Identification of potential biomarkers differentiating MD responses in Low-HF.** Because of its clinical relevance, we decided to focus on analyzing the special case of proto-type-patients in which the treatment reduces HF (Low-HF) but produces MD adverse effect (High-HF). In order to find these prototype-patients, we: (i) identified 13 Low-HF ∩ Low-MD MoAs and 12 Low-HF ∩ High-MD MoAs; and (ii) compared the protein signal of the two groups and proposed 30 potential biomarkers (**Table 1**). Among the proposed biomarkers, we found 16 proteins active in Low-HF ∩ Low-MD MoAs but inactive in Low-HF ∩ High-MD (15 of them shared with MD best-classifier proteins). On the other hand, 14 proteins were identified as inactive in Low-HF ∩ Low-MD and active in Low-HF ∩ High-MD MoAs (12 of them were MD best-classifier proteins). We calculated the GO enriched functions of these two groups and observed that "phosphatidylinositol bisphosphate kinase activity" is enriched among proteins that are active in Low-HF ∩ Low-MD MoAs. Instead, "fibrinolysis" was found to be enriched among proteins active in Low-HF ∩ High-MD MoAs (**Table 2**). With this, we conclude that among the group of prototype-patients for which sacubitril/valsartan improves HF treatment response, the modulation of fibrinolysis could play a role at inducing the MD adverse effect. Moreover, we propose 12 best-classifier proteins that may be considered as bio-markers for good prognosis of the side effect.

In fact, since neovascular MD development is characterized by subretinal extravasations of novel vessels derived from the choroid (CNV) and the subsequent hemorrhage into the photo-receptor cell layer in the macula region [51], it might be reasonable to think that the

**Table 1. Potential biomarker proteins, with opposite signal in Low-HF ∩ Low-MD and Low-HF ∩ High-MD MoAs.**

| | Uniprot ID | Gene symbol | Gene name | ⟨*LMD*⟩ | ⟨*HMD*⟩ | $\sqrt{|LMDxHMD|}$ | Adjusted P-value | BCP |
|---|---|---|---|---|---|---|---|---|
| 1 | **P02675** | **FGB** | **Fibrinogen beta chain** | -0.576 | 0.814 | 0.685 | 1.297E-03 | MD |
| 2 | O43639 | NCK2 | Cytoplasmic protein NCK2 | 0.620 | -0.697 | 0.657 | 1.656E-04 | MD |
| 3 | P54762 | EPHB1 | Ephrin type-B receptor 1 | 0.317 | -0.677 | 0.464 | 3.669E-04 | HF&MD |
| 4 | Q9Y4H2 | IRS2 | Insulin receptor substrate 2 | 0.417 | -0.465 | 0.440 | 8.181E-04 | MD |
| 5 | O60674 | JAK2 | Tyrosine-protein kinase JAK2 | -0.747 | 0.249 | 0.431 | 1.656E-04 | MD |
| 6 | P06241 | FYN | Tyrosine-protein kinase Fyn | 0.591 | -0.236 | 0.373 | 2.466E-04 | HF&MD |
| 7 | P30530 | AXL | Tyrosine-protein kinase receptor UFO | 0.392 | -0.330 | 0.360 | 2.111E-04 | MD |
| 8 | **Q02297** | **NRG1** | **Pro-neuregulin-1, membrane-bound isoform** | 0.672 | -0.188 | 0.355 | 2.111E-04 | MD |
| 9 | P32004 | L1CAM | Neural cell adhesion molecule L1 | -0.373 | 0.309 | 0.339 | 1.297E-03 | HF&MD |
| 10 | Q05586 | GRIN1 | Glutamate receptor ionotropic, NMDA 1 | -0.174 | 0.620 | 0.329 | 1.955E-04 | MD |
| 11 | **P05230** | **FGF1** | **Fibroblast growth factor 1** | -0.152 | 0.688 | 0.323 | 8.181E-04 | HF&MD |
| 12 | **P18084** | **ITGB5** | **Integrin beta-5** | 0.436 | -0.236 | 0.321 | 2.111E-04 | MD |
| 13 | **P01583** | **IL1A** | **Interleukin-1 alpha** | 0.174 | -0.472 | 0.287 | 1.955E-04 | MD |
| 14 | P10275 | AR | Androgen receptor | 0.349 | -0.201 | 0.265 | 8.008E-04 | MD |
| 15 | P15941 | MUC1 | Mucin-1 subunit alpha | 0.099 | -0.652 | 0.254 | 6.905E-04 | HF&MD |
| 16 | O14757 | CHEK1 | Serine/threonine-protein kinase Chk1 | 0.436 | -0.142 | 0.248 | 1.549E-03 | MD |
| 17 | P15391 | CD19 | B-lymphocyte antigen CD19 | -0.131 | 0.357 | 0.216 | 8.160E-03 | MD |
| 18 | **P61981** | **YWHAG** | **14-3-3 protein gamma, N-terminally processed** | 0.174 | -0.236 | 0.203 | 2.783E-03 | - |
| 19 | Q9Y478 | PRKAB1 | 5'-AMP-activated protein kinase subunit beta-1 | 0.261 | -0.142 | 0.192 | 5.682E-03 | MD |
| 20 | P62158 | CALM1; CALM2; CALM3 | Calmodulin-1 {ECO:0000312|HGNC:HGNC:1442} | -0.282 | 0.107 | 0.174 | 9.405E-03 | MD |
| 21 | **P06748** | **NPM1** | **Nucleophosmin** | 0.261 | -0.107 | 0.167 | 3.618E-03 | MD |
| 22 | O15357 | INPPL1 | Phosphatidylinositol 3,4,5-trisphosphate 5-phosphatase 2 | -0.261 | 0.094 | 0.157 | 3.618E-03 | MD |
| 23 | P17081 | RHOQ | Rho-related GTP-binding protein RhoQ | -0.218 | 0.094 | 0.143 | 9.794E-03 | MD |
| 24 | P35354 | PTGS2 | Prostaglandin G/H synthase 2 | 0.044 | -0.472 | 0.143 | 3.669E-04 | MD |
| 25 | P42684 | ABL2 | Abelson tyrosine-protein kinase 2 | -0.218 | 0.094 | 0.143 | 9.794E-03 | MD |
| 26 | **Q15109** | **AGER** | **Advanced glycosylation end product-specific receptor** | -0.267 | 0.063 | 0.130 | 8.160E-03 | - |
| 27 | P07585 | DCN | Decorin | -0.044 | 0.236 | 0.101 | 5.682E-03 | MD |
| 28 | **P05155** | **SERPING1** | **Plasma protease C1 inhibitor** | -0.044 | 0.236 | 0.101 | 5.682E-03 | MD |
| 29 | **P05121** | **SERPINE1** | **Plasminogen activator inhibitor 1** | -0.044 | 0.236 | 0.101 | 5.682E-03 | - |
| 30 | P14770 | GP9 | Platelet glycoprotein IX | 0.044 | -0.236 | 0.101 | 5.682E-03 | MD |

Highlighted cells correspond to proteins that are part of the Top-HF ∪ Top-MD ∪ Top-Drug set, the top-scoring proteins according to GUILDify. Columns show: the protein name (as UniprotID, gene-symbol and gene-name), the average of the signal in in Low-MD (<LMD>) and High-MD (<HMD>) in the selected sets of MoAs and a measure of the strength of the signal in both distributions (calculated as $\sqrt{LMDxHMD}$), the significance (adjusted P-value) ensuring that both distributions of signals are different, and whether the protein has been considered best-classifier in MD of HF (BCP).

modulation of fibrinolysis and blood coagulation pathways could play a role. The reported implication of some fibrinolysis related classifiers, such as FGB, SERPINE1 (PAI-1), and SERPING1, in neovascular MD development seems to support this hypothesis [52–54]. Besides, valsartan might be implicated in this mechanism, since it has been reported to modulate PAI-1 levels and promote fibrinolysis in different animal and human models [55,56]. In addition, the presence of several other MD related classifiers in this list, such as IRS2 [57], PTGS2 [58], DCN [59] and FGF1 [60], further supports the interest of the classifiers as biomarkers of MD development in sacubitril/valsartan good responders. Still, we would like to

**Table 2. Top 10 Gene Ontology functions enriched from proteins with opposite signal in Low-HF ∩ Low-MD and Low-HF ∩ High-MD MoAs.**

| | Low-HF ∩ LMD+ HMD- | | | Low-HF ∩ HMD+ LMD- | | | Overlapped functions | | |
|---|---|---|---|---|---|---|---|---|---|
| | GO name | LOD | P-val. | GO name | LOD | P-val. | GO name | LOD | P-val. |
| 1 | phosphatidylinositol-4,5-bisphosphate 3-kinase activity | 1.89 | 0.03600 | fibrinolysis | 2.51 | 0.00050 | response to stimulus | 1.19 | <0.00050 |
| 2 | cellular response to UV | 1.87 | 0.04200 | negative regulation of wound healing | 2.13 | 0.00050 | positive regulation of transport | 1.24 | <0.00050 |
| 3 | phosphatidylinositol bisphosphate kinase activity | 1.87 | 0.04200 | negative regulation of blood coagulation | 2.12 | 0.00850 | positive regulation of biological process | 1.13 | 0.00051 |
| 4 | vascular endothelial growth factor receptor signaling pathway | 1.86 | 0.04200 | negative regulation of hemostasis | 2.12 | 0.00850 | positive regulation of developmental process | 1.18 | <0.00050 |
| 5 | positive regulation of protein kinase B signaling | 1.70 | 0.01050 | negative regulation of coagulation | 2.10 | 0.01050 | positive regulation of cellular process | 1.04 | 0.00294 |
| 6 | negative regulation of apoptotic signaling pathway | 1.68 | 0.00050 | platelet alpha granule lumen | 1.96 | 0.02300 | positive regulation of response to stimulus | 1.04 | 0.00417 |
| 7 | peptidyl-tyrosine phosphorylation | 1.63 | 0.01400 | regulation of epithelial cell apoptotic process | 1.96 | 0.02300 | - | - | - |
| 8 | regulation of apoptotic signaling pathway | 1.63 | <0.00050 | regulation of blood coagulation | 1.91 | 0.02800 | - | - | - |
| 9 | peptidyl-tyrosine modification | 1.62 | 0.01400 | regulation of hemostasis | 1.91 | 0.02800 | - | - | - |
| 10 | protein tyrosine kinase activity | 1.61 | 0.01850 | regulation of coagulation | 1.89 | 0.03450 | - | - | - |

Functional enrichment analysis from FuncAssociate [50].

highlight that the biomarkers have been proposed using a theoretical approach, and that the clinical effects studied may not be present in real patients.

## 4. Analysis of proposed biomarkers with GUILDify

In the previous section, we proposed 30 proteins that could potentially help to identify HF patients at risk of developing MD. To corroborate these biomarkers, we tested how many of them are found using a different approach also based on the use of functional networks. For this purpose, we used GUILDify v2.0 [23], a web server that extends the information of disease-gene associations through the protein-protein interactions network. GUILDify scores proteins according to their proximity with the genes associated with a disease (seeds). Using this web server, we identify a list of top-scoring proteins that are critical on transmitting the perturbation of disease genes through the network. The network used by GUILDify is completely independent from the HPN used in the TPMS, becoming an ideal, independent context to test the potential biomarkers.

Thus, we used GUILDify to indicate which of the potential biomarkers identified by TPMS may have a relevant role in the molecular mechanism of the drug. We ran GUILDify using the two targets of sacubitril/valsartan (NEP, AT1R) as seeds, and selected the top 2% scored nodes (defined as the "top-drug" set). We did the same with the phenotypes of HF and MD, using as seeds the 124 effectors of HF and 163 effectors of MD from the BED database. We merged the top scored sets of HF, MD and top-drug ("top-drug ∪ top-HF ∪ top-MD") and studied the overlap with the set of 30 biomarkers proposed in the previous section. 10 of the candidate biomarkers are found in the merged set "top-drug ∪ top-HF ∪ top-MD" and are consequently significant (see **S1 File**).

Some of these candidates can be functionally linked to both diseases and the drug under study. For example, among these 10 classifiers, AGER has been implicated in both HF [61], through extracellular matrix remodeling, and MD development [62], through inflammation,

oxidative stress, and basal laminar deposit formation between retinal pigment epithelium cells and the basal membrane; furthermore, this receptor is known to be modulated by AT1R [63], valsartan target. Similarly, FGF1 has been proposed to improve cardiac function after HF [64], as well as to promote choroid neovascularization leading to MD [47]. Moreover, FGF1 is regulated by angiotensin II through ATGR2 [65], another protein suggested as classifier in the current analysis that is known to mediate some of the effects of AT1R antagonists, such as valsartan [41,42]. Another candidate, NRG1, has been linked to myocardial regeneration after HF [66] and is known to lessen the development of neurodegenerative diseases such as Alzheimer's disease [67], which shares similar pathological features with MD [68]. NRG1 is also linked to the expression of neprilysin [67], sacubitril target. ITGB5 has been identified as risk locus for HF [69] and its modulation has been linked to lipofucsin accumulation in MD [70]. Interestingly, ATGR1 inhibitors have been reported to modulate ITGB5 expression in animal models [71]. Finally, IL1A has been proposed as an essential mediator of HF pathogenesis [72,73] through inflammation modulations, and serum levels of this protein have been found increased in MD patients [74]. In addition, as described in previous sections, classifiers FGB, SERPINE1, and SERPING1 have been linked to MD [52–54] and are also known to play a role in HF development [75–78]. According to these findings, the 10 potential biomarkers proposed by TPMS and identified with GUILDify might be prioritized when studying good responder HF patients at risk of MD development.

## Limitations

Although TPMS returns the amount of signal from the drug arriving to the rest of the proteins in the HPN, this signal is only a qualitative measure. We are not using data about the dosage of the drug or the quantity of expression of the proteins. However, we are already working to make TPMS move towards the growing tendency of Quantitative Systems Pharmacology. The quantification of the availability of drugs in the target tissue for each patient opens the opportunity to have an accurate patient simulation to do *in silico* clinical trials.

## Conclusions

It exists an increasing need for new tools to get closer to real life clinical problems and the Systems Biology-based computational methods could be the solution needed. The specific case of sacubitril/valsartan stands out because of the amount of resources invested in the safety of the drug and the concern on the possible risk of inducing amyloid accumulation-associated conditions, such as macular degeneration (MD), in the long term. In this study, we applied TPMS technology to uncover different Mechanisms of Action (MoAs) of sacubitril/valsartan over heart failure (HF) and reveal its molecular relationship with MD. For this approach, we hypothesize that each MoA would correspond to a prototype-patient. The method is then used to generate a wide battery of MoAs by performing an *in silico* trial of the drug and pathology under study. TPMS computes the models by using a hand curated Human Protein Network and applying a Multilayer Perceptron-like and sampling method strategy to find all plausible solutions. After analyzing the models generated, we found different sets of proteins able to classify the models according to HF treatment efficacy or MD treatment relationship. The sets include functions such as PI3K and MAPK kinase signaling pathways, involved in HF-related cardiac hypertrophy, or fibrinolysis and coagulation processes (e.g. FGB, SERPINE1 or SERPING1) and growth factors (e.g. FGF1 or PDGF) related to MD induction. Furthermore, we propose 30 biomarker candidates to identify patients potentially developing MD under a successful treatment with sacubitril/valsartan. Out of this 30, 10 biomarkers were also found in the alternative, independent molecular context proposed by GUILDify, including some HF

and MD effectors such as AGER, NRG1, ITGB5 or IL1A. Further studies might prospectively validate the herein raised hypothesis.

We notice that the models generated with TPMS are completely theoretical and thus, they are not associated with clinical effects of real patients. Consequently, the biomarkers proposed on the basis of these models are also theoretical and would require an experimental validation. Still, TPMS represents a huge improvement for studying the hypothetical relationship between a drug and an adverse effect. Until now, there were not enough tools that allow to perform an exhaustive study on the MoAs of an adverse effect. Now, with the MoAs and biomarkers proposed by TPMS, we provide the tools for this type of research.

## Supporting information

**S1 File. Extended version of materials and methods; S1-S5 Figs; S1-S13 Tables.** (DOCX)

## Author Contributions

**Conceptualization:** Narcís Fernández-Fuentes, José Manuel Mas, Baldo Oliva.

**Data curation:** Guillem Jorba, Valentin Junet, Cristina Segú-Vergés, José Luis Ruiz, José Manuel Mas.

**Formal analysis:** Guillem Jorba, Joaquim Aguirre-Plans, Cristina Segú-Vergés, Albert Pujol, Narcís Fernández-Fuentes, José Manuel Mas, Baldo Oliva.

**Funding acquisition:** José Manuel Mas.

**Investigation:** Guillem Jorba, Joaquim Aguirre-Plans, Cristina Segú-Vergés, Narcís Fernández-Fuentes, José Manuel Mas, Baldo Oliva.

**Methodology:** Guillem Jorba, Joaquim Aguirre-Plans, Albert Pujol, Narcís Fernández-Fuentes, José Manuel Mas, Baldo Oliva.

**Resources:** José Manuel Mas.

**Software:** Guillem Jorba, Joaquim Aguirre-Plans, Albert Pujol, José Manuel Mas.

**Supervision:** José Manuel Mas, Baldo Oliva.

**Validation:** Guillem Jorba, Joaquim Aguirre-Plans, José Manuel Mas, Baldo Oliva.

**Visualization:** Guillem Jorba, Joaquim Aguirre-Plans.

**Writing – original draft:** Guillem Jorba, Joaquim Aguirre-Plans, Cristina Segú-Vergés, Narcís Fernández-Fuentes.

**Writing – review & editing:** Guillem Jorba, Joaquim Aguirre-Plans, Valentin Junet, Cristina Segú-Vergés, José Luis Ruiz, Albert Pujol, Narcís Fernández-Fuentes, José Manuel Mas, Baldo Oliva.

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
