## [Decision Letter · Decision Letter 0]

29 Oct 2019

PONE-D-19-21700

TPMS technology to infer biomarkers of macular degeneration prognosis in in silico simulated prototype-patients under the study of heart failure treatment with sacubitril and valsartan

PLOS ONE

Dear Prof. Oliva,

Thank you for submitting your manuscript to PLOS ONE. After careful consideration, we feel that it has merit but does not fully meet PLOS ONE’s publication criteria as it currently stands. Therefore, we invite you to submit a revised version of the manuscript that addresses the points raised during the review process.

Additionally to the reviewers comments, I would like to ask the authors to more clearly highlight the theoretical nature of their work that addresses mainly the theoretical possibility of the method but not necessarily clinical effects. This is particularly true in view of the absence of evidence that the side effect investigated is relevant in patients. The authors should interpret their findings with these thoughts in mind. Finally, the authors should give some more insight in how they think the method should be used for future research and how this may influence both research and clinical practice.

We would appreciate receiving your revised manuscript by Dec 13 2019 11:59PM. To enhance the reproducibility of your results, we recommend that if applicable you deposit your laboratory protocols in protocols.io, where a protocol can be assigned its own identifier (DOI) such that it can be cited independently in the future. For instructions see: http://journals.plos.org/plosone/s/submission-guidelines#loc-laboratory-protocols

We look forward to receiving your revised manuscript.

Kind regards,

Hans-Peter Brunner-La Rocca, M.D.

Academic Editor

PLOS ONE

Journal Requirements:

2. Thank you for stating the following in the Financial Disclosure section:"BO is awarded by the Spanish Ministry of Economy (MINECO) with grant BI2017-85329-R The funders had no role in study design, data collection and analysis, decision to publish, or preparation of the manuscript."

Thank you for stating the following in the Competing Interests section:"I have read the journal's policy and the authors of this manuscript have the following competing interests: Baldo Oliva, currently serves on the editorial board as academic editor of PLOS ONE."

We note that one or more of the authors are employed by a commercial company: "Anaxomics Biotech SL, Barcelona"

b) . Please also provide an updated Competing Interests Statement declaring this commercial affiliation along with any other relevant declarations relating to employment, consultancy, patents, products in development, or marketed products, etc. 

Reviewers' comments:

Reviewer's Responses to Questions

**Comments to the Author**

1. Is the manuscript technically sound, and do the data support the conclusions?

Reviewer #1: Yes

Reviewer #2: Yes

2. Has the statistical analysis been performed appropriately and rigorously? 

Reviewer #1: I Don't Know

Reviewer #2: Yes

3. Have the authors made all data underlying the findings in their manuscript fully available?

Reviewer #1: Yes

Reviewer #2: Yes

4. Is the manuscript presented in an intelligible fashion and written in standard English?

Reviewer #1: Yes

Reviewer #2: Yes

5. Review Comments to the Author

Reviewer #1: The Authors applied the Therapeutic Performance Mapping System (TPMS) technology to the prediction of macular degeneration (MD) in patients receiving sacubitril valsartan. They report that "a lower response in term of heart failure treatment is more associated to macular degeneration development" and propose "a set of 30 potential biomarkers... to identify mechanisms (or patients) more prone to suffering macular degeneration when presenting good heart failure response".

The primary targets of this paper are data scientists or information engineers. As a clinical cardiologist, I refrain from judging the technical aspects of the study. I just make a comment about the study design and the plausibility of results.

The Authors present MD as "a common/recurrent adverse effect" of therapy with sacubitril valsartan, which does not seem to be the case. Indeed, 5 years after the publication of the PARADIGM HF study, 3 years after the latest ESC guidelines and 2 years after the update to ACC/AHA guidelines, a dedicated literature search does not give any results except for some outdated concerns of a greater risk of Alzheimer's disease (AD) and MD based on conceptual considerations. See for example doi 10.1038/nrcardio.2016.200: "Additionally, inhibition of neprilysin metabolism of amyloid-β peptides might have an effect on Alzheimer disease, age-related macular degeneration, and cerebral amyloid angiopathy". Nonetheless, the increased risk of AD has not been confirmed by dedicated studies. Therefore, the Authors propose a sophisticated approach inevitably burdened by a lot of assumptions and simplifications to solve a problem (i.e., how to predict MD) that does not seem to exist.

Reviewer #2: Jorba et al., used the Therapeutic Performance Mapping System (TPMS) approach to look for biomarkers which can predict MD in HF patients treated with sacubitril/valsartan. The potential for in-silico clinical trials is clearly shown by this system biology approach. There is a strong methodological base to test the hypothesis on. Although it is based on a string of assumptions inherent to the methodology, which make the eventual translation difficult. At some points the manuscript seems a bit tedious, and some analyses seem to be redundant.

Major

- The approach depends heavily on assumptions and definitions. For example, the HPN is created based on proteins related to the disease (heart failure in this case) from BED. It is difficult to retrieve these proteins used as input, however this input of proteins determines strongly all the other analyses. HF is an extreme heterogeneous disease which has many etiologies and a diverse scale on pathomechanisms. For example: is the input from BED mainly based on ischemic HF, or HF due to abnormal loading conditions?

- Is tissue-specificity taking into account? Restrictions of the HPN are based on gene expression datasets. However, many proteins/genes involved in the pathogenesis of HF are tissue-specific to the heart. Eventually there is a link in the lowHF and highMD group pointing towards fibrinolysis, how should I interpret these results in the light of tissue specificity?

- In line with previous point, in the end a list of biomarkers is proposed, the best-classifier proteins can be used as biomarker. Although no suggestions are made how these should reach clinical implementation. Should these markers be measured in blood, or are they only measurable as RNA in cardiac biopsies?

- How are the MD effectors determined? Are they also retrieved from BED? The HPN is build upon the BED input from HF. How complete is this one for MD?

- It’s difficult to interpret the TSignal values (supplemental figure 1). Low and high is defined as first and fourth quartile. How do the 2nd and 3th quartile fit in supplementary figure 1. How is the distinguishment among the four quartiles, or is there an overlapping spectrum from first to fourth quartile?

- It is a bit confusing that certain aspects seem to intertwine. To me, the most interesting part is the analysis regarding high versus low MD in the lowHF group. However, before this analysis there is a lot of emphasis on the low versus high HF group, which is also interesting, but seems not be the purpose of this manuscript and the further analysis.

- In addition, it is not clear why the last analysis using GUILDify is performed. Also in the conclusion it is stated that 30 biomarkers are proposed (out of the previous analysis). But thereafter, 10 out of the 30 are proposed to be involved in the comorbidity between HF and MD. What does this mean? The multiple analyses seem to introduce more confusion than clarity at this point.

Minor

- Try to focus the introduction immediately towards HF instead of cardiovascular diseases in general.

- Figure 1 (especially a) is difficult to read. I can’t read the text in this figure.

- In figure 2, how can there be BCP in the upper right corner? What does a differential non-best classifier protein exactly mean?

- What does a differential non-BCP imply?

- Are there more proteins which are differential best-classifier proteins which did not reach significance? Ie Opposite effect in low versus high, although not p<0.01.

- Input for the GO enriched function LHF+HHF- has only 6 proteins as input, this seems a very low input for a GO enrichment analysis.

- The approach with the Hausdorff and Euclidean distances seems a bit redundant as the MDS plot better shows how it ‘actually’ works. Visually, the graph is a bit unattractive, as there is much overlap between the groups, indicating that there is no clear clustering. Maybe the graph has to be split into a high and low HF plot, to better show how the MD groups cluster within the HF groups.

- The part that describes how biomarkers are selected is to confusing (352-370), try to describe it more compact and to the point.

- Line 416-17 “In Fig 2, the differencial best-classifier proteins with higher score can be identified by a larger area”. It is not clear what is meant with this sentence?

- Line 462-64 “We found that… to MD”. This sentence does not make sense and is not in line with the rest of the manuscript.

6. PLOS authors have the option to publish the peer review history of their article (what does this mean?). If published, this will include your full peer review and any attached files.

Reviewer #1: No

Reviewer #2: No

---

## [Author Response · Author response to Decision Letter 0]

20 Dec 2019

Funding Statement

GJ, VJ, CSV, JLR, AP and JMM have commercial affiliation to Anaxomics Biotech SL. The funders provided support in the form of salaries for authors JAP, NFF, BO, GJ and VJ, but did not have any additional role in the study design, data collection and analysis, decision to publish, or preparation of the manuscript. The specific roles of these authors are articulated in the ‘author contributions’ section.

Competing Interests Statement

The commercial affiliation of the authors does not alter our adherence to PLOS ONE policies on sharing data and materials

Response to reviewers: Please see attached cover letter

---

## [Decision Letter · Decision Letter 1]

13 Jan 2020

PONE-D-19-21700R1

In-silico simulated prototype-patients using TPMS technology to study a potential adverse effect of sacubitril and valsartan

PLOS ONE

Dear Prof. Oliva,

Thank you for submitting your manuscript to PLOS ONE. After careful consideration, we feel that it has merit but does not fully meet PLOS ONE’s publication criteria as it currently stands. Therefore, we invite you to submit a revised version of the manuscript that addresses the points raised during the review process.

ACADEMIC EDITOR: 

Both reviewers were satisfied with the reply and the changes made. However, you did not address my comment, which has been as follows:

Additionally to the reviewers comments, I would like to ask the authors to more clearly highlight the theoretical nature of their work that addresses mainly the theoretical possibility of the method but not necessarily clinical effects. This is particularly true in view of the absence of evidence that the side effect investigated is relevant in patients. The authors should interpret their findings with these thoughts in mind. Finally, the authors should give some more insight in how they think the method should be used for future research and how this may influence both research and clinical practice.

Some of the changes made partly address this but not completely. I would like to ask you to address these points specifically and adjust the manuscript accordingly.

We would appreciate receiving your revised manuscript by Feb 27 2020 11:59PM. To enhance the reproducibility of your results, we recommend that if applicable you deposit your laboratory protocols in protocols.io, where a protocol can be assigned its own identifier (DOI) such that it can be cited independently in the future. For instructions see: http://journals.plos.org/plosone/s/submission-guidelines#loc-laboratory-protocols

We look forward to receiving your revised manuscript.

Kind regards,

Hans-Peter Brunner-La Rocca, M.D.

Academic Editor

PLOS ONE

Reviewers' comments:

Reviewer's Responses to Questions

**Comments to the Author**

1. If the authors have adequately addressed your comments raised in a previous round of review and you feel that this manuscript is now acceptable for publication, you may indicate that here to bypass the “Comments to the Author” section, enter your conflict of interest statement in the “Confidential to Editor” section, and submit your "Accept" recommendation.

Reviewer #1: All comments have been addressed

Reviewer #2: All comments have been addressed

2. Is the manuscript technically sound, and do the data support the conclusions?

Reviewer #1: Yes

Reviewer #2: Yes

3. Has the statistical analysis been performed appropriately and rigorously? 

Reviewer #1: Yes

Reviewer #2: Yes

4. Have the authors made all data underlying the findings in their manuscript fully available?

Reviewer #1: Yes

Reviewer #2: Yes

5. Is the manuscript presented in an intelligible fashion and written in standard English?

Reviewer #1: Yes

Reviewer #2: Yes

6. Review Comments to the Author

Reviewer #1: The Authors have modified their paper to address the issues raised by the Reviewers. I have no further comments.

Reviewer #2: Thank you for the elaborate reponse to my questions. I feel that all my comments have been addressed accordingly.

7. PLOS authors have the option to publish the peer review history of their article (what does this mean?). If published, this will include your full peer review and any attached files.

Reviewer #1: No

Reviewer #2: No

---

## [Author Response · Author response to Decision Letter 1]

23 Jan 2020

Dear Editor,

We wish to thank you for giving us a new opportunity of revising the text (manuscript reference PONE-D-19-21700R1) and apologize if the previous review was not more explicitly highlighting that the approach was theoretical. Please, find our answers to your comments and the modified manuscript files enclosed. Modifications to the last submitted version of the manuscript are highlighted in green background and modifications to the first revision in yellow. In the present review, we have reinforced the theoretical nature of our work that does not involve necessarily clinical effects. We hope the minor amendments herein introduced will make the current manuscript version suitable for publication. 

Yours sincerely,

José Manuel Mas and Baldo Oliva

Anaxomics Biotech SL

Structural Bioinformatics Group (Universitat Pompeu Fabra)

Funding Statement

GJ, VJ, CSV, JLR, AP and JMM have commercial affiliation to Anaxomics Biotech SL. The funders provided support in the form of salaries for authors JAP, NFF, BO, GJ and VJ, but did not have any additional role in the study design, data collection and analysis, decision to publish, or preparation of the manuscript. The specific roles of these authors are articulated in the ‘author contributions’ section.

Competing Interests Statement

The commercial affiliation of the authors does not alter our adherence to PLOS ONE policies on sharing data and materials

 

ACADEMIC EDITOR:

Both reviewers were satisfied with the reply and the changes made. However, you did not address my comment, which has been as follows:

Additionally to the reviewers comments, I would like to ask the authors to more clearly highlight the theoretical nature of their work that addresses mainly the theoretical possibility of the method but not necessarily clinical effects. This is particularly true in view of the absence of evidence that the side effect investigated is relevant in patients. The authors should interpret their findings with these thoughts in mind. Finally, the authors should give some more insight in how they think the method should be used for future research and how this may influence both research and clinical practice.

Some of the changes made partly address this but not completely. I would like to ask you to address these points specifically and adjust the manuscript accordingly.

As the editor suggested, we have emphasized more clearly the theoretical nature of our work. We remarked the theoretical nature of the results in both the abstract and the introduction:

All prototype-patients models generated are completely theoretical and therefore they do not necessarily involve clinical effects in real patients. 

[…]

In this study, we used TPMS and GUILDify v2.0 to analyze the relationship between sacubitril/valsartan, HF and MD in entirely theoretical models. Because these are theoretical models it is important to note that they are not associated with clinical effects in real patients, they only point on potential mechanisms to explain potential adverse effects.

Additionally, we also highlighted that the prototype-patient models are theoretical in the section “2. Comparison of MoAs with high/low TSignal associated to HF or MD” of the Results and discussion:

Finally, we highlight that, as these distinct groups of prototype-patients are theoretical simulations, they don’t reflect the clinical effects of real patients.

We also remarked the theoretical nature of the biomarkers in the section “3. Identification and functional analysis of potential biomarkers” of the Results and Discussion:

Still, we would like to highlight that the biomarkers have been proposed using a theoretical approach, and that the clinical effects studied may not be present in real patients.

Finally, we included in the Conclusions a discussion about the theoretical nature of the results and the future influence of the method in research and clinical practice:

We notice that the models generated with TPMS are completely theoretical and thus, they are not associated with clinical effects of real patients. Consequently, the biomarkers proposed on the basis of these models are also theoretical and would require an experimental validation. Still, TPMS represents a huge improvement for studying the hypothetical relationship between a drug and an adverse effect. Until now, there were not enough tools that allow to perform an exhaustive study on the MoAs of an adverse effect. Now, with the MoAs and biomarkers proposed by TPMS, we provide the tools for this type of research.

---

## [Editor Report · Decision Letter 2]

28 Jan 2020

In-silico simulated prototype-patients using TPMS technology to study a potential adverse effect of sacubitril and valsartan

PONE-D-19-21700R2

Dear Dr. Oliva,

We are pleased to inform you that your manuscript has been judged scientifically suitable for publication and will be formally accepted for publication once it complies with all outstanding technical requirements.

With kind regards,

Hans-Peter Brunner-La Rocca, M.D.

Academic Editor

PLOS ONE

---

## [Editor Report · Acceptance letter]

29 Jan 2020

PONE-D-19-21700R2 

In-silico simulated prototype-patients using TPMS technology to study a potential adverse effect of sacubitril and valsartan 

Dear Dr. Oliva:

I am pleased to inform you that your manuscript has been deemed suitable for publication in PLOS ONE. Congratulations! Your manuscript is now with our production department. 

With kind regards,

on behalf of

Dr. Hans-Peter Brunner-La Rocca 

Academic Editor

PLOS ONE